# No Critical Ultrasound, No Life: The Value of Point-of Care Critical Ultrasound in the Rescue of Critically Ill Infants

**DOI:** 10.3390/diagnostics13243624

**Published:** 2023-12-08

**Authors:** Jing Liu, Ya-Li Guo, Xiao-Ling Ren

**Affiliations:** 1Department of Neonatology and NICU, Beijing Obstetrics and Gynecology Hospital, Capital Medical University, Beijing 100026, China; 2Department of Neonatology and NICU, Beijing Chao-Yang Hospital, Capital Medical University, Beijing 100043, China; 3Department of Neonatology and NICU, Beijing Chao-Yang District Maternal and Child Health Care Hospital, Beijing 100021, China

**Keywords:** point-of-care critical ultrasound (POC-CUS), lung ultrasound, critical disease, newborn infant

## Abstract

Point-of-care critical ultrasound (POC-CUS) screening plays an increasingly important role in the treatment of critically ill infants. Without POC-CUS, the lives of many infants would not be saved in time and correctly. A premature infant with systemic multiple organ system dysfunction caused by fungal sepsis was treated and nursed under the guidance of POC-CUS monitoring, and the infant was ultimately cured. This premature infant had systemic multiple organ system dysfunction and disseminated intravascular coagulation (DIC) caused by fungal sepsis. In the hypercoagulable state of early-stage DIC, cardiac thrombosis could be found using ultrasound screening. For this case, right renal artery thrombosis was found via renal artery Doppler ultrasound examination. Due to the severity of this disease, ultrasound-guided peripherally inserted central catheter (PICC) insertion and ultrasound checks of the PICC tip’s position were performed, which ensured the success of this one-time catheterization and shortened the catheterization time. Lung ultrasound is used for the diagnosis and differential diagnosis of pulmonary diseases, and to guide the application of mechanical ventilation. Because the abdominal circumference of the patient’s markedly enlarged abdominal circumference, bloody stool, and absence of bowel sounds, abdominal ultrasonography was performed, which revealed a markedly enlarged liver, significant peritoneal effusion, and necrotizing enterocolitis. Guided by POC-CUS monitoring, we had the opportunity to implement timely and effective treatment that ultimately saved this critically ill patient’s life. The successful treatment of this newborn infant fully reflects the importance of carrying out POC-CUS screening.

Thrombosis is a rare disease in the neonatal period, which is mainly seen in children with PICCs [1,2,3,4,5]. The preterm infant studied in this paper had sepsis and DIC due to fungal infection. Point-of-care (POC) echocardiography was performed due to increased heart rate and cardiac dysfunction; the results showed a hyperechoic mass with a size of 0.60 cm × 0.45 cm between the left coronary valve of the aorta and the aortic wall, suggesting a cardiac thrombus (Figure 1A). Because this premature patient also had hypertension and hematuria, color Doppler ultrasound examination of the renal artery was performed, which showed that the systolic velocity of the left renal artery was 87.4 cm/s, the end-diastolic velocity was 24.5 cm/s, and the resistance index was 0.72, indicating that the hemodynamic parameters of the left renal artery were normal (Figure 1B). However, the blood flow spectrum of the right renal artery showed that the systolic blood flow velocity decreased and the diastolic blood flow presented with reversed perfusion, that is, the infant had a significant hemodynamic disorder in her right renal artery. Combined with the coexisting hypertension and hematuria, the findings suggested that the patient had right renal artery thrombosis (Figure 1C). No cases of both cardiac and renal artery thrombosis have been reported in the literature to date [1,2,3,4,5].

A female newborn infant, G_1_P_1_, vaginally delivered at 34^+2^ weeks of gestational age and with a birth weight of 1200 g (below the third percentile of 1540 g), was admitted to the Neonatal Intensive Care Unit (NICU) 20 min after birth. The pregnant mother had a 40 h premature rupture of membranes with severe preeclampsia and fungal vaginitis. The patient’s Apgar score was 10-10-10/1-5-10 min. After admission, the physical examination, arterial blood gas analysis, routine blood tests, and coagulation function tests all gave normal results. Lung ultrasound (LUS) showed bilateral confluent B-lines, alveolar-interstitial syndrome, and subpleural consolidation with air bronchograms in some intercostal spaces, consistent with ultrasound image characteristics of neonatal pneumonia [6,7,8] (Figure 2A). The brain ultrasound images and cerebral hemodynamic monitoring were normal (Figure 3). Therefore, the patient was diagnosed with preterm infant, very low birth weight infant, and small for her gestational age, and with neonatal pneumonia (mild). Cefoperazone sulbactam sodium was given to prevent bacterial infections, and fluconazole was given to prevent fungal infections. Ultrasound-guided peripherally inserted central catheter (PICC) insertion was performed due to the low birth weight and estimated long hospital stay, and the post-catheterization ultrasound positioning showed an accurate PICC tip position [9,10,11] (Figure 4). Twenty-four hours after admission, cardiac ultrasound examination suggested patent ductus arteriosus (PDA), and one course of oral ibuprofen was administered to treat the PDA. Until Day 13 after birth, the patient’s condition was generally stable, with various laboratory tests showing normal results and no bacterial growth in the blood cultures.

However, on Day 14 after birth, the patient’s condition began to deteriorate suddenly, with the patient developing a fever with a skin temperature of 39 °C. Routine blood tests showed a white blood cell count (WBC) of 10.54 × 10^9^/L, a platelet count (PLT) of 233 × 10^9^/L; CRP 1.43 mg/L, and a PCT of 0.47 ng/mL (please see the dynamic changes in routine blood tests and infection indicators in Table 1). The patient had an abnormal coagulation function: APTT 53.6 s, TT 19.6 s, Fig 2.43 g/L, DD 9.41 mg/L, and FDPs 22.9 mg/L (please see the dynamic changes in coagulation function in Table 2). Her antibiotics were upgraded to meropenem to enhance anti-infective treatment.

On Day 16 after birth, the patient still had a fever, with a highest skin temperature of 38.2 °C. Routine blood examination showed that her WBC had decreased to 4.78 × 10^9^/L, PLT had decreased to 130 × 10^9^/L, CRP had increased to 9.0 mg/L, and PCT was 0.66 ng/mL. The patient’s coagulation function further deteriorated: APTT, PT, and TT could not be detected, and the Fig decreased to 1.28 g/L, DD decreased to 4.2 mg/L, and FDPs decreased to 14.4 mg/L. *Candida albicans* was isolated from both the blood and PICC tip position. The additional diagnoses were neonatal fungal sepsis and disseminated intravascular coagulation (DIC). The treatment was as follows: (1) intensified antifungal therapy, in which fluconazole was replaced by caspofungin to enhance antifungal treatment; (2) de-escalation of antibiotics; and (3) heparin and low-molecular weight dextran for anti-DIC therapy, tranexamic acid for anti-fibrinolytic therapy, and fibrinogen and plasma supplements for symptomatic treatment.

On Day 18 after admission, the patient’s condition still showed a trend of aggravation, and her fever was still 38.8 °C. Routine blood test results were still abnormal, the WBC had decreased to 4.78 × 10^9^/L, the PLT had decreased to 29 × 10^9^/L, the CRP had increased to 97.1 mg/L, and the PCT had increased to 11.1 ng/mL. The results of coagulation function tests were as follows: PT 23.5 s, APTT 64.8 s, Fig 2.35 g/L, TT 32.6 s, DD 26.3 mg/L, and FDPs 92 mg/L. After withdrawal from the high-level antibiotics, the infant’s temperature continued to rise (38.2–38.8 °C) instead of falling. Meropenem was given again for anti-infective treatment, and the body temperature decreased. Although the patient’s body temperature now showed a downward trend, she developed severe edema, oliguria, gross hematuria and proteinuria, an increased blood pressure (97/75 mmHg), severe dyspnea, and an increased heart rate, to >180–200/min. Her blood biochemistry was as follows: TBIL 51.1 µmol/L, DBIL 29.3 µmol/L, IBIL 21.8 µmol/L, AST 23 U/L, ALT 71 U/L, ALB 38 g/L to 25.1 g/L, CK-MB 117 U/L, BUN 12.7 mmol/L, and CRE 131 µmol/L. Routine urinalysis showed a cloudy appearance with protein and red blood cells. Microscopic examination showed white blood cells and red blood cells. Renal hemodynamic monitoring showed right renal artery diastolic retrograde perfusion (Figure 1B,C). LUS showed bilateral lung consolidation with air bronchogram signs, especially in the right lung, with regular boundaries. Due to the patient’s history of fungal infection, fungal pneumonia and atelectasis were considered [6,12] (Figure 2B). Echocardiography revealed a normal cardiac structure and function and a normal pulmonary artery pressure. The supplementary diagnoses were as follows: renal artery embolism and neonatal pneumonia with atelectasis (fungal). On the basis of anti-infection, anti-DIC, and anti-fibrinolytic therapy, as described above, urokinase thrombolytic therapy and invasive ventilator-assisted respiration were assigned to this patient.

However, the patient’s condition still showed a trend of deterioration. On Day 20 after birth, her generalized edema worsened, her abdomen was highly distended, her abdominal ultrasound varied significantly, her liver was significantly enlarged into the pelvis, and her bowel sounds had disappeared. After the body temperature had dropped, the heart rate continued to increase, and a grade III systolic murmur was heard in the precordial area. An abdominal ultrasound showed liver enlargement and peritoneal effusion, suggesting necrotizing enterocolitis (NEC) [13,14,15] (Figure 5), and a cardiac ultrasound showed a hyperechoic mass between the left coronary valve of the aorta and the aortic wall, suggesting a cardiac thrombus (Figure 1A). Combined with an increased level of direct bilirubin (DBIL 29.3 µmol/L) and abnormal liver function (AST 923 U/L, ALT 71 U/L), the patient’s ALB decreased from 38.3 to 25.1 g/L, and the myocardial enzymes were elevated (CK-MB117 U/L). A reexamination of the infection indicators found significantly abnormal results: CRP 97.1–107.7 mg/L, PCT 11.1–13.41 ng/mL; fecal occult blood (+). There was also a severe decrease in fibrinogen; even under the condition of daily supplementation of fibrinogen combined with anti-fibrinolytic therapy, the monitored figure was still <0.63 g/L daily (lower than the instrument can detect). The supplementary diagnoses were as follows: cardiac thrombosis, capillary leak syndrome (CLS), NEC, ascites, and hepatic dysfunction. 

Treatment measures: On the basis of anti-infection treatment, ventilator treatment, and anti-DIC treatment, treatments such as thrombolytic, anti-fibrinolytic, and fibrinogen supplementation, 3% NaCl, and multiple abdominal punctures for drainage, among others, were continued. The results after three days of thrombolytic therapy showed that the gross hematuria disappeared and the renal artery blood flow had returned to normal. Five days later, the cardiac emboli had shrunk. After seven days of invasive ventilator use, an LUS showed that the left lung was completely normal, and the pleural line and A-line were clearly presented, the pleural line of the right lung was blurred, and a few B-lines still remained; therefore, the criteria of indications for ultrasound-guided ventilator withdrawal formulated by our team were met, and the ventilator was safely removed [16] (Figure 2C).

The patient’s temperature returned to normal 26 days after birth (10 days after intensified antifungal therapy). The ascites disappeared, and the liver size returned to normal 27 days after birth. The coagulation function returned to normal after 10 days of anti-DIC treatment. At 30 days after birth, the patient’s physical symptoms and various indicators tended to be stable. She was discharged after 58 days of hospitalization, at which time her brain ultrasound, MRI, and electroencephalogram were all normal. No further complications, such as bronchopulmonary dysplasia (BPD) or retinopathy of prematurity (ROP), occurred. After 32 months of follow-up monitoring, the patient’s growth and development improved.

Point-of-care critical ultrasound (POC-CUS) has played increasingly important roles in clinical critical care management and has been widely used in the fields of adults, children, and newborns [17,18,19,20,21,22]. It can even be said that without POC-CUS, the lives of many newborn infants would be difficult to save. In this late-preterm infant, fungal sepsis caused a series of systemic severe diseases during hospitalization, and the management of these was guided by POC-CUS monitoring, which ultimately saved the infant’s life. On the basis of her premature birth and low birth weight, the patient acquired fungal sepsis during vaginal delivery, which resulted in multiple organ failure, disseminated intravascular coagulation, renal artery embolism, cardiac thrombosis, CLS, NEC, fungal pneumonia, and other complications. After active and correct treatment, the critically ill patient was finally cured. During hospitalization, the newborn infant underwent cranial ultrasound, cardiac ultrasound, lung ultrasound, abdominal ultrasound, and Doppler ultrasound screening because of her condition, which excluded and helped to clarify the existing complications of various organs and systems, especially her cardiac embolism, renal artery thrombosis, and the changes in her pulmonary diseases, and played an important role in guiding the correct treatment of the disease. It can be said that without the guidance of routine bedside critical ultrasound screening, the prognosis of this child would be difficult to estimate.

## Figures and Tables

**Figure 1 diagnostics-13-03624-f001:**
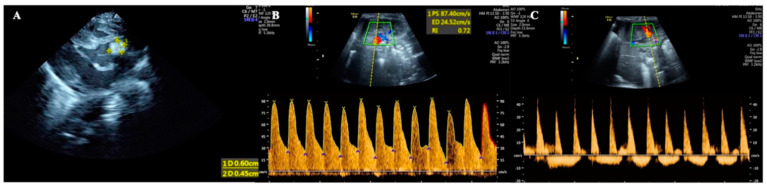
Cardiac and renal artery thrombus. (**A**) POC echocardiography showed a cardiac thrombus with a size of 0.60 cm × 0.45 cm between the left coronary valve of the aorta and the aortic wall. (**B**) Renal artery color Doppler ultrasound showed that the normal Hemodynamic spectrum in the left kidney. (**C**) Renal artery color Doppler ultrasound showed that reversed perfusion in the right kidney.

**Figure 2 diagnostics-13-03624-f002:**
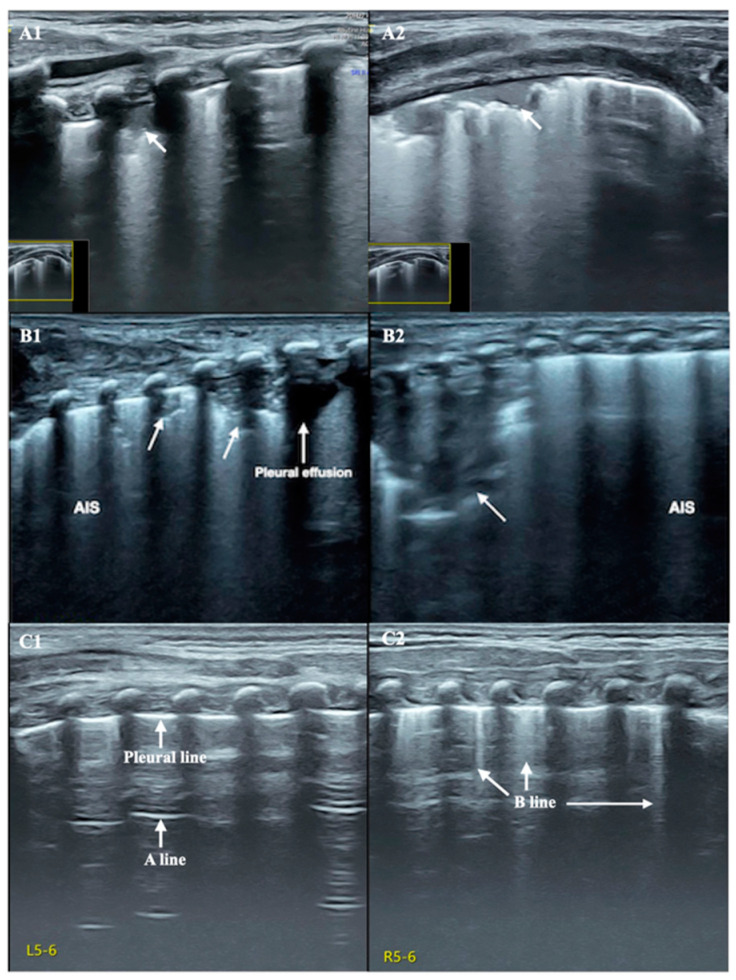
Lung ultrasound. (**A**) LUS revealed consolidation involving one intercostal space with irregular borders, without air bronchogram signs (arrow) ((**A1**) perpendicular scanning; (**A2**) parallel scanning). (**B**) LUS showed bilateral lungs ((**B1**) left lung, (**B2**) right lung) consolidation involving multiple intercostal spaces with irregular borders (arrows), especially in the right lung, and alveolar-interstitial syndrome (AIS), suggesting pneumonia. (**C**) LUS showed that a pleural line and A-line clearly existed in the left lung field (**C1**), and that there were a few B-lines in the right lung (**C2**), which met the requirements for weaning under ultrasound monitoring proposed by our team, thus the patient could be safely weaned from the ventilator.

**Figure 3 diagnostics-13-03624-f003:**
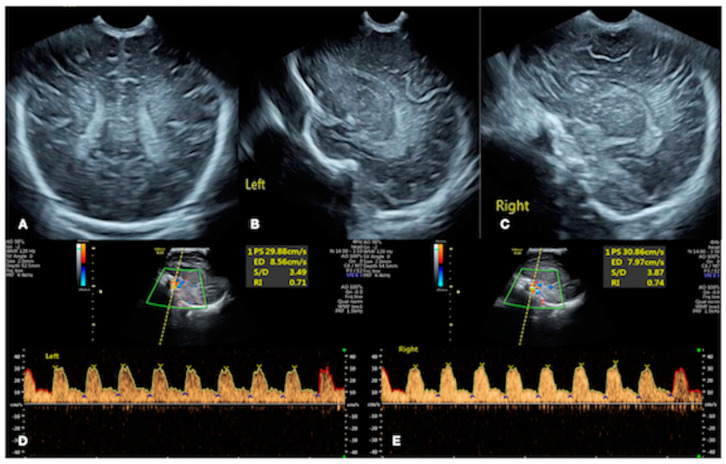
Brain ultrasound. Cranial ultrasound showed a normal echo in the brain parenchyma, centered in the midline of the brain, a regular shape of the choroid plexus, and no abnormal hy-perechoic mass in the brain parenchyma and ventricles ((**A**) coronal section; (**B**) left sagittal section; (**C**) right sagittal section). Color Doppler ultrasound showed normal hemodynamic parameters, such as the flow velocity and resistance index of bilateral anterior cerebral artery ((**D**) left anterior cerebral artery; (**E**) right anterior cerebral artery).

**Figure 4 diagnostics-13-03624-f004:**
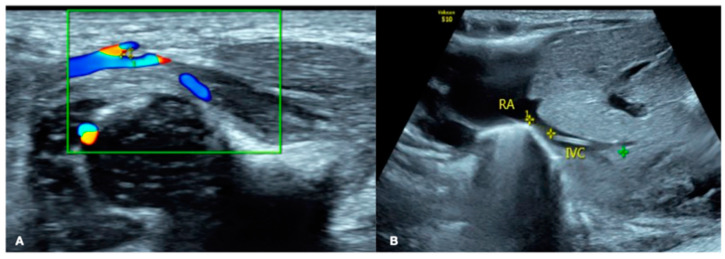
Ultrasound-guided PICC placement. (**A**) First step, color Doppler ultrasound was used to check the course of the great saphenous vein (GSV) and the angle between the GSV and femoral vein (FV) to determine which side of the blood vessel was the easiest for placement. (**B**) Second step, then, after successful catheterization, ultrasound was used to determine whether the position of the PICC tip was accurate.

**Figure 5 diagnostics-13-03624-f005:**
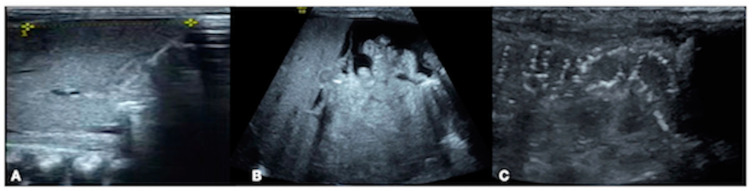
Ultrasound of the abdomen. Abdominal ultrasound showed that the liver was significantly enlarged to reach the pelvic cavity (**A**) and ascites (**B**), and (**C**) showed significant pneumatosis intestinalis (punctate or linear hyperechogenicity), indicating the presence of NEC.

**Table 1 diagnostics-13-03624-t001:** Dynamic changes in routine blood test results and infection indicators.

Testing Date after Birth	WBC (×10^9^/L)	PLT (×10^9^/L)	CRP (mg/L)	PCT (ng/mL)
Day 14	10.54	233	1.43	0.47
Day 16	4.78	130	9.0	0.66
Day 18	4.78	29	97.1	11.1
Day 20	-	-	107.7	13.41

Note: Twenty days after birth, the above indicators gradually returned to normal levels.

**Table 2 diagnostics-13-03624-t002:** Dynamic changes in coagulation function.

Testing Date after Birth	APTT (s)	PT (s)	TT (s)	Fig (g/L)	DD (mg/L)	FDPs (mg/L)
Day 14	53.6	19.6	22.7	2.43	9.41	22.9
Day 16	>	>	>	1.28	4.2	14.4
Day 18	64.8	23.5	32.6	2.35	26.3	92.0
Day 20	-	-	-	<	-	-

Note: (1) >: Exceeding the detection limit; <: below the detection limit; -: no record. (2) After 24 days of age, the patient’s coagulation function gradually returned to normal.

## Data Availability

The data used and analyzed in this study are available from the corresponding author upon reasonable request.

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
