# Peer review of "No Critical Ultrasound, No Life: The Value of Point-of Care Critical Ultrasound in the Rescue of Critically Ill Infants"

_diagnostics, 2023, doi:10.3390/diagnostics13243624_

Round 1

Reviewer 1 Report

Comments and Suggestions for Authors

The authors report a case of cardiac and renal thrombosis which was successfully managed by including point-of-care ultrasound.

The title is a bit exaggerated – I would omit the first part.

The first half of the Abstract needs some corrections – of the English language, of the tense used (it should be past tense) and also, the fourth sentence should be deleted. Also, on the third line, saving lives is not a thing that is performed ”correctly” – this also should be deleted.

I wish the authors used the CARE algorithm to construct their case presentation, things would have been much clearer then. Even using the classic sections of a manuscript (Introduction, Case Report, Discussions, Conclusions) would have led to a better organization of their work. Also, I’m not sure why they chose to delete line numbers on the side of the page for review purposes...

The first paragraph in the main body of the manuscript is not meant to be placed there, it should be moved onwards.

On the second paragraph the authors mention neonatal pneumonia solely based on the ultrasound examination. Did the infant not present ANY physical signs of respiratory distress and no need for respiratory support? On the same paragraph, the authors should delete the sentence mentioning the diagnoses, especially since those are not relevant to the rest of the infant’s outcome.

On the fourth page, the authors mentioning upgrading the antibiotic regimen to meropenem – that implies the infant was on antibiotic prophylaxis for 14 days with cefoperazone sulbactam? There is no mention of lumbar puncture, either…

On page 5, fibrinogen is “lower than the instrument can detect” – I would rephrase this as “below detection limit”.

In the end, I feel the authors try to overplay the role of ultrasound in “saving the infant’s life”. Nothing else is mentioned, there is no comparison with similar cases or studies. Where are the proper discussions for this case?

On page 6, the list is missing some abbreviations used, e.g. TT, BUN…

Comments on the Quality of English Language

Must be improved

Author Response

Response to Reviewer 

The authors report a case of cardiac and renal thrombosis which was successfully managed by including point-of-care ultrasound.

The title is a bit exaggerated – I would omit the first part.

Response: We agree with the suggestion and have made the changes as requested.

The first half of the Abstract needs some corrections – of the English language, of the tense used (it should be past tense) and also, the fourth sentence should be deleted. Also, on the third line, saving lives is not a thing that is performed ” correctly” this also should be deleted.

Response: We revised the abstract section accordingly and invited the native English speaker to revise the language throughout the paper.

I wish the authors used the CARE algorithm to construct their case presentation, things would have been much clearer then. Even using the classic sections of a manuscript (Introduction, Case Report, Discussions, Conclusions) would have led to a better organization of their work. Also, I’m not sure why they chose to delete line numbers on the side of the page for review purposes...

Response: Dear reviewer, this article was written in accordance with the requirements and format of Interesting Images and not in the format of a case report. Therefore, we did not revise the format of the article during the revision.

The first paragraph in the main body of the manuscript is not meant to be placed there, it should be moved onwards. 

Response: Dear reviewer, due to the same reason, this paragraph was not moved, please refer to the above.

On the second paragraph the authors mention neonatal pneumonia solely based on the ultrasound examination. Did the infant not present ANY physical signs of respiratory distress and no need for respiratory support? On the same paragraph, the authors should delete the sentence mentioning the diagnoses, especially since those are not relevant to the rest of the infant’s outcome.

Response: On admission, the patient had very mild pneumonia on ultrasound, which was not sufficient to cause significant dyspnea. Therefore, the infant did not have obvious respiratory distress on physical examination on admission, therefore, mechanical ventilation is not required. This also reflects one of the advantages of ultrasound, which can detect very mild lung lesions with great sensitivity. Please note that the diagnoses were deleted in the revised edition.

On the fourth page, the authors mentioning upgrading the antibiotic regimen to meropenem – that implies the infant was on antibiotic prophylaxis for 14 days with cefoperazone sulbactam? There is no mention of lumbar puncture, either…

Response: Dear reviewer, thank you very much for your preciseness and seriousness. Early antibiotic treatment was used for one week. Given the severity of the infection at this stage, intensive antibiotic therapy was administered. Please note that we have revised these contents (including the corresponding part of the second paragraph).

On page 5, fibrinogen is “lower than the instrument can detect” – I would rephrase this as “below detection limit”.

Response: Thank you very much, please note that we have made this change.

In the end, I feel the authors try to overplay the role of ultrasound in “saving the infant’s life”. Nothing else is mentioned, there is no comparison with similar cases or studies. Where are the proper discussions for this case?

Response: Thank you very much. Some sentences suspected of exaggerating have been removed, such as “If It can even be said that without POC-CUS, the lives of many newborn infants will be difficult to save” and “It can be said that without routine bedside critical ultrasound guidance, the prognosis of this child would be difficult to assess timely and accurately” in the last paragraph.

On page 6, the list is missing some abbreviations used, e.g. TT, BUN…

Response: These contents were supplemented and revised.

Reviewer 2 Report

Comments and Suggestions for Authors

The article is not well written. I suggest to improve the article dividing it in introduction, case report and discussion. I suggest to summarize the case report in a table. Moreover, I suggest to reduce the not relevant clinical data. Images are well chosen.

Author Response

Response to Reviewer 

The article is not well written. I suggest to improve the article dividing it in introduction, case report and discussion. I suggest to summarize the case report in a table. Moreover, I suggest to reduce the not relevant clinical data. Images are well chosen.

Response: Dear reviewer, this article was written in accordance with the requirements and format of Interesting Imageswhile not by the case report, therefore, the format of the article could not be written in accordance with the format of a case report. Interesting Images has its own format requirements.

These clinical data including laboratory test results help to reflect the severity of the disease. Therefore, if deleted, it may not reflect the severity of the infant's disease. Point-of-care critical ultrasound plays an important role in finding the changes of the condition timely,  making accurate judgment timely, and taking effective treatment timely.

Round 2

Reviewer 1 Report

Comments and Suggestions for Authors

Thank you to the authors for their willingness to implement the changes suggested. The manuscript is closer to a publishable form. There are still some minor issues:

-          On page 3 in this new version, it is said that the infant “was admitted to the hospital 20 minutes after birth”. I think this should be reformulated as “was admitted to the Neonatal Intensive Care Unit 20 minutes after birth”, since she was probably born in the hospital.

-          On page 7 and onwards, I would point out that the measured temperature was central or skin, as the case may be.

-          Again, on page 7, the first value of CRP was 14.3 mg/L. However, in the next paragraph, the value is said to be “increased” to 9 mg/L – please, make the necessary changes. Also, correct the value of PCT to ng/mL, instead of ng/ml, in both paragraphs.

-          Still on page 7, it is said that “APTT and TT could not be detected, and they decreased to 1.28 g/L” – I suspect that “they” is actually fibrinogen – please, make the necessary changes.

-          For ease of reading, I suggest placing the biochemical and hematological values in a table and referring to that throughout the text.

-          Regarding the References, for a more uniform appearance, I suggest replacing https://doi.org/10..... with DOI (in capitals): 10….

Comments on the Quality of English Language

Minor issues

Author Response

Thank you to the authors for their willingness to implement the changes suggested. The manuscript is closer to a publishable form. There are still some minor issues:

-          On page 3 in this new version, it is said that the infant “was admitted to the hospital 20 minutes after birth”. I think this should be reformulated as “was admitted to the Neonatal Intensive Care Unit 20 minutes after birth”, since she was probably born in the hospital.

Response: Yes,it is. We have revised it in the revision.

-          On page 7 and onwards, I would point out that the measured temperature was central or skin, as the case may be.

Response: It’s skin temperature, we modified it.

-          Again, on page 7, the first value of CRP was 14.3 mg/L. However, in the next paragraph, the value is said to be “increased” to 9 mg/L – please, make the necessary changes. Also, correct the value of PCT to ng/mL, instead of ng/ml, in both paragraphs.

Response: Thanks for so carefully, it was 1.43 mg/L at the beginning. The ng/mL has been corrected.

-          Still on page 7, it is said that “APTT and TT could not be detected, and they decreased to 1.28 g/L” – I suspect that “they” is actually fibrinogen – please, make the necessary changes.

Response: Thank you very much. It’s fibrinogen.

-          For ease of reading, I suggest placing the biochemical and hematological values in a table and referring to that throughout the text.

Response: We have added Table I and Table, which shows the changes in blood routine test and coagulation function during the various periods.

-          Regarding the References, for a more uniform appearance, I suggest replacing https://doi.org/10..... with DOI (in capitals): 10….

Response: we have modified them.